# Effectiveness of a Real-Time Ventilation Feedback Device for Guiding Adequate Minute Ventilation: A Manikin Simulation Study

**DOI:** 10.3390/medicina56060278

**Published:** 2020-06-05

**Authors:** Sejin Heo, Sun Young Yoon, Jongchul Kim, Hye Seung Kim, Kyunga Kim, Hee Yoon, Sung Yeon Hwang, Won Chul Cha, Taerim Kim

**Affiliations:** 1Department of Emergency Medicine, Samsung Medical Center, Sungkyunkwan University School of Medicine, Seoul 06351, Korea; silversh06@naver.com (S.H.); roseherb21@naver.com (S.Y.Y.); wildhi.yoon@samsung.com (H.Y.); sygood.hwang@samsung.com (S.Y.H.); wc.cha@samsung.com (W.C.C.); 2Department of Biomedical Engineering, Samsung Medical Center, Seoul 06351, Korea; goodman0707.kim@samsung.com; 3Biostatistics and Clinical Epidemiology Center, Samsung Medical Center, Seoul 06351, Korea; hyeseung.kim@sbri.co.kr (H.S.K.); Kyunga.j.kim@samsung.com (K.K.); 4Department of Digital Health, Samsung Advanced Institute for Health Sciences & Technology, Sungkyunkwan University, Seoul 06355, Korea

**Keywords:** positive pressure ventilation, bag valve, cardiopulmonary resuscitation, simulation study

## Abstract

*Background and objectives:* It is often challenging even for skilled rescuers to provide adequate positive pressure ventilation consistently. This study aimed to investigate the effectiveness of a newly developed real-time ventilation feedback device (RTVFD) that estimates tidal volume (TV) and ventilation interval (VI) in real time. *Materials and methods:* We conducted a randomised, crossover, manikin simulation study. A total of 26 medical providers were randomly assigned to the RTVFD-assisted ventilation (RAV) first group (*n* = 13) and the non-assisted ventilation (NV) first group (*n* = 13). Participants provided ventilation using adult and paediatric bag valves (BVs) for 2 min each. After a washout period, the simulation was repeated by exchanging the participants’ groups. *Results:* The primary outcome was optimal TV in the RAV and NV groups using adult and paediatric BVs. A secondary outcome was optimal VI in the RAV and NV groups using adult and paediatric BVs. The proportions of optimal TV values were higher for the RAVs when using both adult and paediatric BVs (adult BV: 47.29% vs. 18.46%, *p* < 0.001; paediatric BV: 89.51% vs. 72.66%, *p* < 0.001) than for the NVs. The proportions of optimal VI were significantly higher in RAVs when using both adult and paediatric BVs than that in NVs (adult BV: 95.64% vs. 50.20%, *p* < 0.001; paediatric BV: 95.83% vs. 57.14%, *p* < 0.001). Additionally, we found that with paediatric BVs, the simulation had a higher OR for both optimal TV (13.26; 95% CI, 9.96–17.65; *p* < 0.001) and VI (1.32; 1.08–1.62, *p* = 0.007), regardless of RTVFD use. *Conclusion:* Real-time feedback using RTVFD significantly improves the TV and VI in both adult and paediatric BVs in a manikin simulation study.

## 1. Introduction

Despite the continuous efforts to improve the survival rate of cardiac arrest patients, the rate remains poor worldwide [1,2,3,4]. Prompt delivery of high-quality cardiopulmonary resuscitation (CPR) ensures a significant survival benefit, according to previous studies [5,6]. The American Heart Association (AHA) and European Resuscitation Council guidelines recommend the performance of chest compression at an adequate rate and depth, with minimal pauses between compressions and with adequate ventilation, namely, optimal tidal volume (TV) and ventilation interval (VI) to produce a visible chest rise [7,8,9,10].

The risks of hyperventilation during CPR, such as decreased cardiac output with increased intrathoracic pressure and decreased cerebral blood due to reduced partial pressure of carbon dioxide in the blood, are well known [11,12,13,14]. Thus, the AHA guidelines for CPR recommend an optimal ventilation rate of 10/min and TV of 6–7 mL/kg in both adult and paediatric patients [10,15]. However, it is often challenging for rescuers to consistently provide adequate positive ventilation using bag valves (BVs) [16,17,18]. **Hyperventilation, even with professional rescuers, has been reported [1,13,14,19]**. The simple manipulating method of BV does not guarantee consistently adequate TV [20].

To avoid suboptimal ventilation, some ancillary devices have been developed and studied; the use of a metronome improved the proportions of correct chest compression and ventilation rates, but it could not monitor TV during CPR [21]. A few studies have shown that paediatric BVs could maintain an acceptable level of oxygenation and suggested that adult BVs should be replaced with paediatric BVs during training to prevent hyperventilation [22,23]. The use of an impedance threshold device improved the haemodynamics and short-term survival but could not provide information about the actual TV and failed to prolong long-term survival [14,24,25]. Other equipment has been manufactured in recent years [26,27]. One of instrument could simultaneously display TV and a bagging time alarm, but TV was only shown through a bar graph variation and was not displayed as the actual numerical volume [26].

To overcome the limitations of the previous devices, we developed a new real-time ventilation feedback device (RTVFD). It displays numerical TV and a bagging time alarm in real time, thus guiding rescuers to achieve optimal ventilation per minute. This study primarily aimed to investigate whether RTVFD increases the proportion of adequate TV when using BVs of different sizes: adult and paediatric BVs. In addition, we compared the proportion of adequate VI with or without the assistance of RTVFD using adult and paediatric BVs.

## 2. Methods

### 2.1. A Real-Time Ventilation Feedback Device

The experimental device was composed of the main board (control board), flow sensor, pressure sensor, organic light-emitting diode display, Bluetooth module, and power (Figure 1A). A controller based on the Arduino Uno Board was created and modified by partial boards for miniaturisation.

A processor sampled three analogue signals at a sampling frequency of 40 Hz (25 ms) using the Microchip’s ATmega328/P in the process of quantisation and converted analogue to digital at 10 bits. The analogue input value of the processor consisted of a mass flow sensor’s (PMF4103V, POSIFA Microsystems Inc., San Jose, CA, USA output value, a pressure sensor’s (XGZP6847010KPG, CFSensor Inc., Anhui, China) output value, and the internal battery’s (TW632570, 3.7 V, 1200 mAh, The HAN Inc., Seoul, Korea) voltage value.

The processor repeated the measurement of the mass flow sensor’s output value. The air content of the mass flow sensor was measured 70 times (every 25 ms), and the accumulated values were used to obtain an input value of 1.75 s in the controller. Subsequently, the amount of air was computed by utilising the output formula provided by the manufacturer:PMF4103V flow rate = ((Vout − 1 V)/4 v) × full-scale flow rate(1)

When the mass flow sensor detected more than a fixed quantity of air, the processor measured the air content and activated two internal timers. The timer measured the time until the next influx to compute the interval of BV ventilation (interval time). The other timer informed the user of the bagging time in real time using the numeric value and an alarm sound (Figure 1B). Moreover, Bluetooth was embedded to link the device with smart devices that measured bagging pressure and air volume in real time through an exclusive application, and this was provided to the users.

### 2.2. Device Validation

The RTVFD was connected to a mechanical ventilator (Hamilton G5) with a volume-controlled mode (inspiration/expiration ratio of 1:5 and a respiration rate of 10/min) at TVs of 150, 250, 350, 450, and 550 mL. We provided each TV to the RTVFD using an analyser (gas flow analyser, FLUKE Corps, Washington, USA) 25 times. Subsequently, we measured the mean, standard deviation (SD), and the difference in the ratios of TVs between the RTVFD and the analyser.
(2)$$\mathrm{Difference}\;\mathrm{in}\;\mathrm{the}\;\mathrm{ratio}\;\mathrm{of}\;\mathrm{TVs}=\frac{TV\;of\;RTVFD\;-\;TV\;of\;analyser}{TV\;of\;analyser}$$

### 2.3. Study Design

We conducted a randomised, crossover, manikin simulation study for a study period from 1 January 2019 to 1 February 2019 at the emergency department of Samsung Medical Center, a tertiary teaching hospital with 250 emergency visits per day. This study was approved by the institutional review board (IRB) of our institute (IRB number: 2018-05-144) on 14 June 2018.

### 2.4. Participants

Based on the pilot study with and without RTVFD using adult BVs, we calculated a sample size of 26 participants per group using the McNemar test for two paired proportions with 2.5% significance and 80% power. The proportion of discordant pairs was 50%; based on that, the proportion of optimal TV of the RTVFD-assisted ventilation (RAV) was 35%, and that of the non-assisted ventilation (NV) was 85%. We hypothesised that there is no difference in the proportion of optimal TV provided by adult and paediatric BVs. Participants were recruited by means of recruitment information posted on the employee bulletin board or hospital intranet of the study site. All the participants were medical providers such as doctors, nurses, and emergency medical technicians (EMTs) who were Advanced Cardiovascular Life Support (ACLS)-certified and at the time working in emergency or critical care units. Individuals who have a hearing disorder or musculoskeletal disease were excluded.

### 2.5. Study Protocol

All participants were provided 30 minutes of introduction to the study protocol and the optimal ventilation recommended by the AHA guidelines for CPR. In the first trial, a total of 26 volunteers were randomly assigned to the RAV first group (*n* = 13) and the NV first group (*n* = 13). We used the block randomisation method. The RAV first group was asked to perform ventilation as per the RTVFD guideline, and the NV first group followed the AHA guideline. 

The manikin (Ambu^®^Man Advanced-Next Generation) was intubated using a cuffed endotracheal tube (ETT) with a 7.5 mm inner diameter, and the depth of ETT placement was 21 cm. The RTVFD was connected between the BV and endotracheal tube (Figure 1C). We applied two different sizes of BVs (Ambu^®^ Mark IV (reusable resuscitator): Ambu Mark IV Adult and Ambu Mark IV Baby). Participants provided ventilation using adult and paediatric BVs for 2 minutes each, with a 2-min break time in between. We measured TV and VI for a total of 4 minutes. The display of the RTVFD was covered while performing non-feedback-assisted ventilation and uncovered during RTVFD feedback-assisted ventilation. TV and VI data were transferred to the application in a handheld device via the Bluetooth of the RTVFD (Figure 1D). We hypothesised that the advanced airway had been secured. This study focused on ventilations only; therefore, chest compressions were not performed.

The optimal TV was defined as 420–480 mL for adult BVs and 120–180 mL for paediatric BVs. We set up a similar volume interval (60 mL) for adult and paediatric BVs. The maximum volumes delivered for an adult BV and paediatric BV were 1300 and 300 mL, respectively [28]. The optimal VI was defined as a breath every 6–8 s. After a 1-week washout period, the simulation was repeated with the participants crossed over to the other group. The flow diagram for the study is shown in Figure 2.

### 2.6. Outcomes

The primary outcome was the proportion of optimal TV in the RAV and NV groups using adult and paediatric BVs. The secondary outcome was the proportion of optimal VI in the RAV and NV groups using adult and paediatric BVs.

### 2.7. Statistics

The baseline characteristics of the two groups were described. Continuous variables with normal distributions were described as means with SD, and those with non-normal distributions were described as medians with interquartile ranges (IQRs). Categorical variables were presented as frequencies with percentages. Student’s t-test was used to compare the means of two continuous variables with normal distributions, and the Wilcoxon rank-sum test was used to compare the means of two groups of variables not normally distributed. The frequencies of categorical variables were compared using the chi-square test. We used generalised estimating equation (GEE) models, extensions from logistic regression models, to analyse binary response variables that were repeatedly measured across multiple time points. In the GEE models, the response variables were the achievement status of adequate TV and that of VI. The use of RTVFD assistance and BV type were included as independent variables, and the interaction effect was also examined along with their main effects. Subgroup analyses were further conducted for each BV type. Results for the GEE are reported as odds ratios (ORs) with 95% confidence intervals (CIs). Differences were considered significant at a 95% level (*p* < 0.05). Statistical analysis was performed using Statistical Analysis System (SAS) version 9.4 (SAS Institute, Cary, NC) and R 3.5.1 (R Foundation for Statistical Computing).

## 3. Results

### 3.1. Real-Time Ventilation Feedback Device Validation

The mean ± SD values of TV in the RTVFD and the analyser are shown in Table 1 and Figure 3. The SD of TVs in the RTVFD was larger than that of the analyser at all target TVs. No SD exceeded 13.7 and 2.2 mL for the RTVFD and analyser, respectively (Table 1). Among the 125 values of the differences in TVs between the two devices, 109 (87.2%) and 117 (93.6%) were within ±5% and ±10%, respectively.

### 3.2. Baseline Characteristics

Twenty-six participants were enrolled and randomly assigned to the two groups: 19 participants were nurses, 4 were EMTs, and 3 were emergency physicians. Table 2 presents the baseline characteristics of the participants. There were no significant differences between the two groups in terms of age, sex, experience, or job.

### 3.3. Results for the Tidal Volume and Ventilation Interval Performance

The results for the TV and VI performance are shown in Table 3. The proportions of optimal TV values were significantly higher for the RAVs when using both adult and paediatric BVs (adult BV: 47.29% vs. 18.46%, *p* < 0.001; paediatric BV: 89.51% vs. 72.66%, *p* < 0.001) (Figure 4A) than for the NVs.

The mean TV of the RAVs and NVs using adult BVs and paediatric BVs both showed significant differences (adult BV: RAVs, 432.00 (63.93) vs. NVs 392.83 (136.36), *p* < 0.001; paediatric BV: RAVs, 144.84 (23.07) vs. NVs 131.74 (38.78), *p* < 0.001). TV variations were smaller for the RAVs than the NVs.

The proportions of optimal VI were significantly higher in RAVs when using both adult and paediatric BVs than that in NVs (adult BV: 95.64% vs. 50.20%, *p* < 0.001; paediatric BV: 95.83% vs. 57.14%, *p* < 0.001) (Figure 4B). The mean VI of the RAVs and NVs using adult BV showed a significant differences with both adult and paediatric BVs (adult BV: RAVs, 6.84 (1.13) vs. NVs 6.37 (3.17), *p* < 0.001; paediatric BV: RAVs, 6.57 (0.99) vs. NVs 6.29 (2.23), *p* < 0.001). The VI variations of the RAVs were lower than those of NVs.

In the GEE analysis, the OR of optimal TV for the RAVs was 3.90 (95% confidence interval (CI), 2.95–5.15) regardless of the BV type (Table 4). The ORs of the TV were 3.90 (2.95–5.15) and 3.21 (2.30–4.48) for the adult and paediatric BV groups, respectively. The OR of optimal VI for the RAVs was 21.78 (13.71–34.61) regardless of BV type. The ORs of VI were 21.78 (13.71–34.61) and 17.25 (10.80–27.56) for the adult and paediatric BV groups, respectively.

In subgroup analysis, the ORs of ventilations using paediatric BVs were higher (13.26 (9.96–17.65) for TV and 1.32 (1.08–1.62) for VI) than that of adult BVs. We conducted interaction analyses between RTVFD use (with or without) and BV type. The results showed that the effect of RAVs was insignificantly different among the types of BVs with regard to TV and VI.

## 4. Discussion

The present study demonstrated that BV ventilation with RTVFD markedly increased the proportion of optimal TV and VI during manikin simulation. Our device could display the numerical TV repeatedly, even at a small volume for paediatric ventilation, with the alarm for the ventilation cue sounding just in time. This study showed higher ORs for RAVs than for NVs; the OR of optimal VI was higher than that of optimal TV with both adult and paediatric BVs. We set the same optimal volume interval (60 mL); we found that with paediatric BVs, the simulation had a higher OR for both optimal TV and VI, regardless of RTVFD use. 

A BV is generally designed with one-third to half of the bag’s entire volume provided at one grasping [16]. However, even trained rescuers experience difficultly in maintaining adequate ventilation, and several studies in this field have emphasised consistent education and training [29,30]. To increase the optimal ventilation, some feedback devices are manufactured, but most of these devices are not applicable to CPR. However, suggestions regarding the educational effect of ancillary devices have been reported in previous studies [27,31]. Using feedback devices such as RTVFD and choosing a BV of adequate size will effectively guide rescuers to deliver optimal ventilation.

Real-time feedback for TV and VI from our device might be useful for professionals to avoid suboptimal ventilation and high respiratory rate. The embedded Bluetooth facilitated the transfer of information to users through an exclusive application. All volunteers could review their own ventilation records after 2 minutes of ventilation. If this device can help professionals to achieve better ventilation performance, non-medical volunteers might benefit as well, though further studies would have to verify that it is still accurate with non-medical staff.

Some studies have suggested the use of a TV feedback device using a turbine flow meter. However, the rotation of the turbine in high TV persists over 6–10 s, which could result in difficulty providing real-time feedback [32]. Another study suggested a flow meter consisting of a round magnet and a spring. However, the difference ratio of TV between the mechanical ventilator and TV device increases during high-TV ventilation because of mechanical friction between the magnet and inner wall of the main body of the TV device. The device we developed is based on the Arduino Uno Board created and modified by partial boards for miniaturisation using a mass flow sensor. This device could measure a wide range of TV values even in a short duration, keeping consistent the difference TV ratio. It could be superior to other equipment in that it displays numerical TV in real time.

Using an RTVFD allows for the appropriate selection of the BV size for adequate TV for patient age and size. A few studies demonstrated that paediatric BVs could maintain adequate oxygenation in adult patients and proposed using paediatric BVs for training to prevent hyperventilation [22,23]. Providing appropriate and consistent volume delivery to adult patients is possible with a smaller, paediatric-sized BV than with an adult BV [23,33]. When the TV was below the recommended level, the paediatric self-inflatable bag was able to reduce gastric inflation while ensuring sufficient lung ventilation as compared to the adult self-inflatable bag [34]. This means that choosing a mechanically and structurally appropriate BV is important to achieve optimal ventilation. Our novel device would overcome the structural limitation of the BV and facilitate optimal ventilation.

The present study has some limitations. First, in the device validation study, the SD of RTVFD was higher than that of the analyser for all TVs, specifically the SD of RTVFD, which was greater than 10% at 550 mL TV. However, our target was 420–480 mL TV for adult BV simulation, and regardless of the RTVFD use, the mean TVs were less than 450 mL with both BV sizes. The difference in the ratios between the analyser and RTVFD was almost within ±10%. However, at the relatively low TV of the ventilator, the difference in ratios was increased. The device could be improved by enhancing precision at both extremes. Second, we did not perform chest compressions on the manikin; therefore, the variation in airway resistance or compliance during chest compression was not reflected. Further studies are needed to verify the accuracy of RTVFD in real-life situations.

## 5. Conclusions

Real-time feedback using RTVFD improved TV and optimal VI with both adult and paediatric BVs in a manikin simulation study. Our novel device could overcome the structural limitations of the BV and facilitate optimal ventilation.

## Figures and Tables

**Figure 1 medicina-56-00278-f001:**
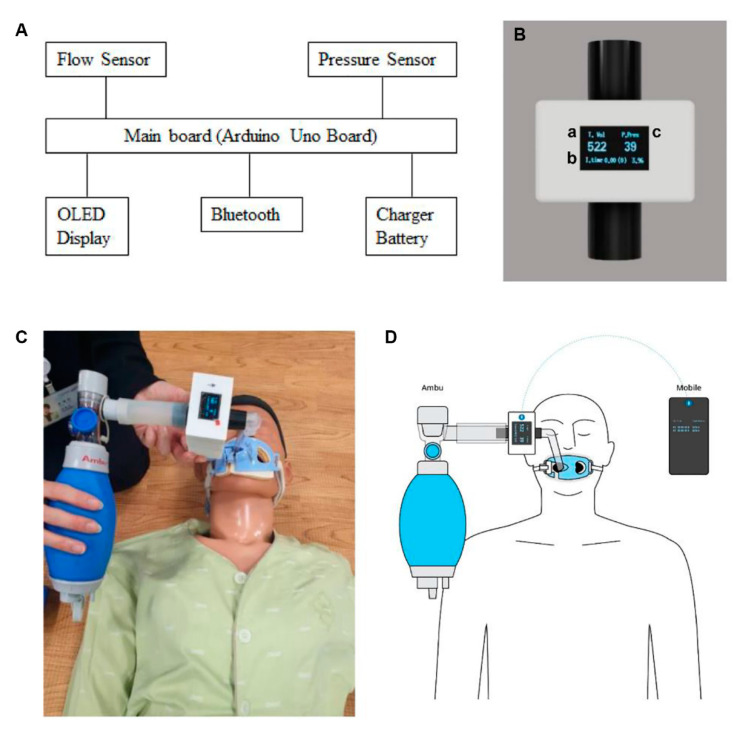
A real-time ventilation feedback device: (**A**) system overview; (**B**) real-time ventilation feedback device (RTVFD)—a: tidal volume, b: inspiration time, counted time right before bagging (seconds to next bagging), residual battery, c: peak pressure; (**C**) paediatric bag valve mask used for manikin simulation; (**D**) schematic diagram of RTVFD setting. RTVFD, real-time ventilation feedback device.

**Figure 2 medicina-56-00278-f002:**
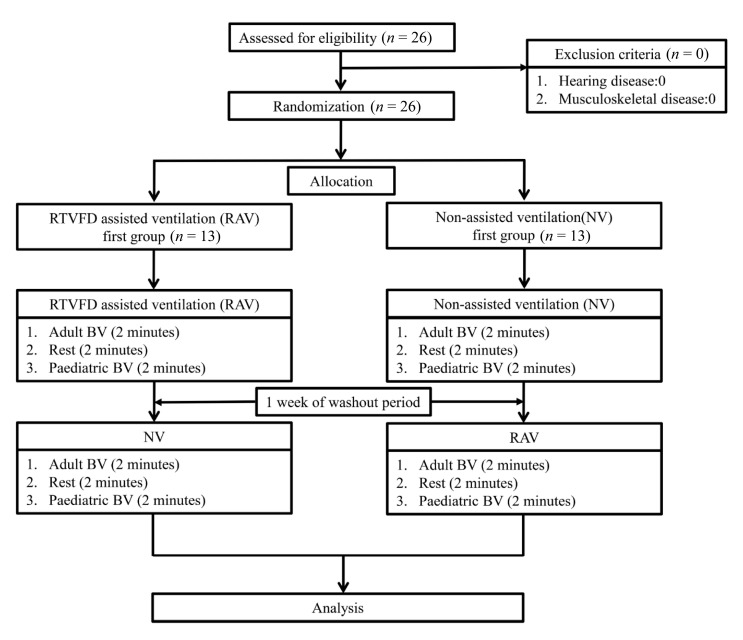
CONSORT flow diagram of the study. RTVFD, real-time ventilation feedback device; BV, bag valve; RAV, real-time ventilation feedback device ventilation; NV, non-assisted ventilation.

**Figure 3 medicina-56-00278-f003:**
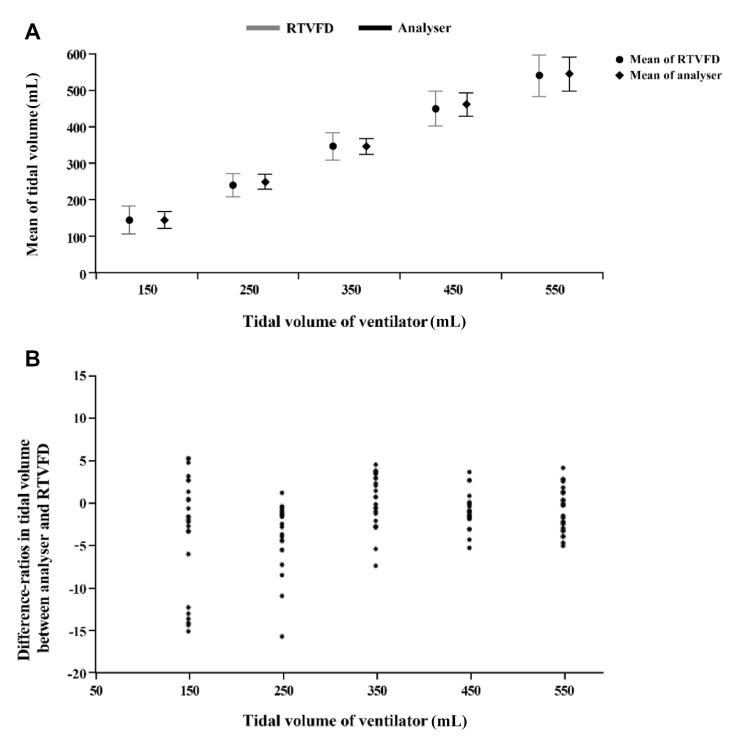
Validation data of RTVFD: (**A**) mean of tidal volume; (**B**) difference ratios in tidal volume between analyser and RTVFD. Difference ratios are defined as the tidal volume of RTVFD-tidal volume of analyser/tidal volume of analyser. RTVFD, real-time ventilation feedback device.

**Figure 4 medicina-56-00278-f004:**
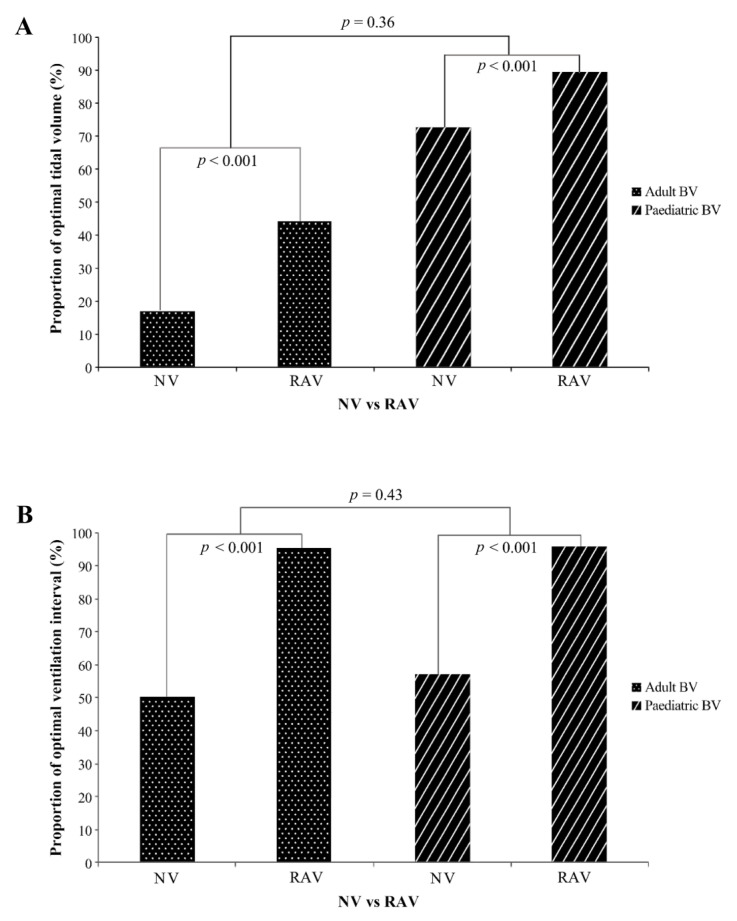
Proportion of optimal TV and VI(%). (**A**) Proportion of optimal TV(%). Comparison of the real-time feedback device-assisted ventilations and the non-assisted ventilations (*n* = 554 for adult, *n* = 553 for paediatric BVs) based on BV type. (**B**) Proportion of optimal VI(%). Non-assisted ventilations (*n* = 506 for adult, *n* = 497 for paediatric BVs) vs. RTVFD-assisted ventilations (*n* = 528 for both adult and paediatric BVs) TV, tidal volume; BV, bag valve; RTVFD, real-time ventilation feedback device; RAV, real-time ventilation feedback device ventilation; NV, non-assisted ventilation.

**Table 1 medicina-56-00278-t001:** Real-time ventilation feedback device validation data.

Set Volume of Mechanical Ventilation, mL	Analyser	RTVFD	*p*-Value
150	149.4 ± 1.0	144 ± 9.8	0.008
250	253.9 ± 0.3	244.3 ± 9.7	<0.001
350	348.2 ± 0.6	348.0 ± 10.9	0.903
450	455.8 ± 1.3	450.1 ± 9.1	0.005
550	551.2 ± 2.2	545.2 ± 13.7	0.040

RTVFD, real-time ventilation feedback device. Values are mean ± SD.

**Table 2 medicina-56-00278-t002:** Baseline characteristics between the two groups.

	RAV First Group (*n* = 13)	NV First Group (*n* = 13)	*p*-Value
Age, median (IQR)	30 (27–33)	28 (26–30)	0.079
Female, N (%)	9 (69.2%)	8 (61.5%)	1.000
Times CPR performed			
Number within 1 week	0.85 (0.80)	1.15 (1.46)	0.512
Number within 1 month	2.15 (2.27)	5.31 (7.04)	0.137
Job, N (%)			0.665
Nurse	10 (76.9)	9 (69.2)	
EMT	2 (15.4)	2 (15.4)	
Doctor	1 (7.7)	2 (15.4)	

RAV, real-time ventilation feedback device ventilation; NV, non-assisted ventilation; IQR, interquartile range; SD, standard deviation; EMT, emergency medical technician.

**Table 3 medicina-56-00278-t003:** The results for the tidal volume and ventilation interval performance.

	Adult BV		*p*-Value	Paediatric BV		*p*-Value
	RAV	NV		RAV	NV	
Tidal volume			<0.001			<0.001
Tidal volume, mL, mean±SD	432.0 ± 63.93	392.83 ± 136.36		144.84 ± 23.07	131.74 ± 38.78	
Optimal tidal volume			<0.001			<0.001
Optimal ventilation, N (%)	262 (47.29)	98 (18.46)		495 (89.51)	380 (72.66)	
Hypoventilation, N (%)	220 (39.71)	309 (58.19)		53 (9.58)	131 (25.05)	
Hyperventilation, N (%)	72 (13.00)	124 (23.35)		5 (0.90)	12 (2,29)	
Ventilation interval			<0.001			<0.001
Ventilation interval, sec	6.84 ± 1.13	6.37 ± 3.17		6.57 ± 0.99	6.29± 2.23	
Optimal interval			<0.001			<0.001
Optimal interval, N (%)	505 (95.64)	254 (50.20)		506 (95.83)	284 (57.14)	

BV, bag valve; RAV, real-time ventilation feedback device ventilation; NV, non-assisted ventilation.

**Table 4 medicina-56-00278-t004:** The results from the generalised estimating equation model for evaluating the use of RTVFD and bag valve type.

	Odds Ratio	95% CI		*p*-Value
**Achievement of adequate TV**				
RTVFD assistance (yes vs. no)	3.90	2.95	5.15	<0.001
BV type (paediatric vs. adult BV)	13.26	9.96	17.65	<0.001
RTVFD assistance (in adult BV subgroup)	3.90	2.95	5.15	<0.001
RTVFD assistance (in paediatric BV subgroup)	3.21	2.30	4.48	<0.001
Interaction (RTVFD assistance & BV type)	0.82	0.54	1.25	0.364
**Achievement of adequate VI**				
RTVFD assistance (yes vs. no)	21.78	13.71	34.61	<0.001
BV type (paediatric vs. adult BV)	1.32	1.08	1.62	0.007
RTVFD assistance (in adult BV subgroup)	21.78	13.71	34.61	<0.001
RTVFD assistance (in paediatric BV subgroup)	17.25	10.80	27.56	<0.001
Interaction (RTVFD assistance & BV type)	0.79	0.44	1.41	0.430

TV, tidal volume; VI, ventilation interval; RTVFD, real-time ventilation feedback device; BV, bag valve; CI, confidence interval; RAV, real-time ventilation feedback device ventilation; NV, non-assisted ventilation.

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
