# Peer review of "Effectiveness of a Real-Time Ventilation Feedback Device for Guiding Adequate Minute Ventilation: A Manikin Simulation Study"

_medicina, 2020, doi:10.3390/medicina56060278_

Round 1

Reviewer 1 Report

Dear authors:

Congratulations for your study and your manuscript. Below my suggestions to improve the clarity of your manuscript:

ABSTRACT: no suggestions

INTRO: no suggestions

METHODS

  • lines 104-105: meaning of sentence 'The proportion of 104 discordant pairs was 50%; based on that, the proportion of optimal TV of the RTVFD-assisted 105 ventilation (RAV) was 35% and that of the non-assisted ventilation (NV) was 85%' i snot clear. Does this refer to discordance between the same participating with and without RTVFD use or discordance between adults and pediatrics? Consider clarify this sentence. IF the former, former, I would move this sentence to the results section.
  • Lines 101-111. Suggest to add where you recruited participants (ER, NICU, PICU, ICUs?) and your randomization method.

RESULTS:

  • Lines 166-170: this information is important for the reader, as it reflects how accurate volumes measured by your RTVFD are compared to the analyzer. Suggest to provide the table in the main manuscript rather than in the supplement. 
  • Line 168: 'No SD exceeded 13.7 and 2.2 for the RTVFD and analyzer, respectively'. Can you add the units (ml, I assume?)?  
  • Line 169-170: 'Among the 125 values of the differences in TVs between the two devices, 109 (87.2%) and 117 (93.6%) were within ±5% and ±10%, respectively'. This sentence is unclear to me. Do you mean that all TV measurements are within 5-10% in each group or between RTVFD vs analyzer?
  • Line 181: consider rephrasing to 'using adult BVs and pediatric BVs both showed significant differences'.
  • Line 182: Consider changing SDs for the term TV variation for more clarity.
  • Line 183-185: consider rephrasing to ' The proportions of optimal TV values were significantly higher for the RAVs when using both adult and paediatric BVs (adult BV: 47.29% vs. 18.46%, p < 0.001; paediatric BV: 89.51% vs 72.66%, p < 0.001) (Figure 3A) than for the NVs'. As this is your primary outcome, suggest to move this sentence to the beginning of the paragraph, prior to specifying the volumes. 
  • Line 186-190: Again, consider moving this sentence with information about your secondary outcome (proportion of adequate VI) to the beginning of the paragraph; add 'significant differences with both adult and pediatric BVs'; change SD for VI variation. 
  • Line 191: 'the OR of optimal TV'.
  • Line 196: does this refer to TV? please, clarify in the text.
  • Lines 197-198: can you add a sentence with information about the significant differences of the interaction analysis? it is unclear to reader which differences you found and therefore, reader cannot figure out whether they are relevant or not.

Table 2: suggest to bold significant p values. Suggest to eliminate* from your legend and add mean+-SD on your first left-side column, same than N (%). 

DISCUSSION (general comment would that your discussion needs some reorganization and editing to highlight the key point of your study)

Line 214: suggest to delete initial sentence and start the paragraph with ' The present study demonstrated that BV ventilation with RTVFD markedly increased the proportion of optimal TV and VI during manikin simulation'. The fact that this may be the first study using RTVFD is less relevant for the reader than the fact that it can improve ventilation performance.

Line 219: suggest change volunteers for professionals. I am not sure I understand what you mean by 'volunteers'. Your study has not looked at results obtained by volunteers. You could argue in a separate sentence that, if this device can help professionals to achieve better ventilation performance, non-medical volunteers might benefit as well, though further studies would have to verify that it is still accurate with non-medical staff.

Lines 222-231: how does this prior evidence compared to your study? It is helpful for the reader to understand the differences between studies and how your study might be potentially more accurate.

Lines 232-235: again, this information refers to your primary and secondary outcomes and therefore should be at the top of the discussion, as part of the first paragraph to facilitate the reader to find the relevant information.

Lines 236-241. This paragraph needs an initial sentence highlighting the key point based on your study results, prior to comparisons with prior literature. For instance, 'using an RTVFD allows appropriate selection of the BV size for adequate TV for patient age and size.'

Lines 244-249: again, this paragraph could be linked to paragraph 232-235 and moved higher up in the text, as this is key information.

Lines 250-267: your limitation paragraph is way too long, almost 40% of your discussion. That raises the question of the validity of the study. Really, your main limitation is the significant differences in volume SD and should be described first. Sample size should not be a limitation as it is in accordance with your sample size estimation and study results are significant. Maybe you can link the fact that this study was done in a manikin with the fact that your participant were within one-year certified life support providers. These are not really  study limitations but a justification of a need for further studies  to verify the accuracy of your device in real life situations. These two concepts can be moved up in the text rather than in the limitation section as well as the opportunity to be part of the education manual. 

Reviewer 2 Report

The authors present a very interesting to investigate the effectiveness of a real-time ventilation feedback device. This is important research because one of the deficiencies in CPR feedback is precise feedback on ventilation rate and tidal volumes. And this could definitely improve delivery of CPR and ultimately OHCA survival rates. I consider this research important. 

In addition, the manuscript is very well written in general, and the results are presented in a clear way. What makes difusion of these important results easier. I think the formatting of the tables should be considerably improved to facilitate understanding of the results. And I have some minor comments/corrections to the manuscript below:

Minor comments

1 Abstract, line 18: remove Background and objectives. The authors have not formatted the abstract as an structured abstract with sections, so this should be removed.

2 Abstract, lines 22-23. Seems like providers were allocated to 1 group only, but in the crossover design the 26 participants did RAV and NV (Figure 2), so please rewrite the abstract mention the crossover design and then have n=26 in both groups.

3 Abstract, in my opinion the main results of the study are the differences in TV (RAV/NV in both groups Adult/Pediatric) and VI. Not the ORs, which I personally find more difficult to interpret. I suggest writtig those resulst (and their p-val) as the main results of the study not on the OR. Or if the authors prefer report and compare the proportions of optimal TV, VI, but not the ORs.

Manuscript:

line 74: 25 mS should be 25 ms.

line 80: “was measured as a datum in every 75ms. Once 70 data were accumulated, these were converted to an input value of” -> was measured 70 times (every 25ms) and the accumulated values were used to obtain an input value of”. Besides the language, I think the authors should explain better what they are doing. I assume flow is measured with a sampling rate of 40s/secm and from there volume measures are taken but it is unclear. Please clarify this section.

lines 94-95: formula, could be put as a formula (in my opinion is more informative than eq (1)) as a LateX equation:

\frac{\text{TV}_\text{RTVFD}-\text{TV}_\text{analyser}}{\text{TV}_\text{analyser}}

More important, the results of callibration are mentioned as a supplementary figure to which I had no access. I think the figure and a description of these results should be part of the main paper. These are important results to characterize the device’s accuracy.

line 106: “provide”-> provided

line 109: use past tense (more occurances in the manuscript): “who are” -> who were (and in line 110 “currently” -> at the time, since maybe now they are not there anylonger).

line 160: “P values less than 0.05” -> Differences were considered significant at a 95% level (p<0.05)” , and use p in lower case as in the tables. In the tables p is italics (math mode), but not here. Be consistent, I would suggest: $\text{p}<0.05$, so avoid italics.

line 162 “(R Foundation fo Statistical Computing, Austria”  -> (R Foundation fo Statistical Computing), close the parenthesis and omitt the country.

line 167 Include Figure S1 and Table S1 as a main results, not supplementary figures/tables. I was not able to find them in the submission but I think they are a main result of the paper indicating how accurate the device is in measuring TV. 

Figure 3a, I cannot understand the upper p-value. From the values displayed, I think there should be significant diferences in optimal TV between pediatric and adult BV, but p=0.3644 is written, odd.  Note use just 2 decimal places for the p-values when not significant.

Tables, I would suggest improving the alignment. For instance in table 1, variables left aligned with headings in bold and then use indentation for the other variables. For instance for the first two columns of table 1

  RAV-first group 
Participants  
   Number 13
   Age 30 (27-33)
   Female (proportion) 9 (69.2%)
Times CPR performed  
   Within a week 0.85 (0.80)
   Within a month 2.15 (2.27)
Job (proportion)  
   Nurse 10 (76.9%)
   .... .....

And the same for Table 2 (use bold face consistently, and separation lines too).

And for Table 3 (bold face Achievement of adequate VI) and then put the variables (with indentation and left aligned as I described for table I) under each analysis
